# Enabling the Phronetically Enacted Self: A Path toward Spiritual Knowledge Management

**Markus F. Peschl** [1,†] , **Alexander Kaiser** [2,*,†] and **Birgit Fordinal** [2,†]

1 Department of Philosophy and Vienna Cognitive Science Hub, University of Vienna, 1010 Vienna, Austria; franz-markus.peschl@univie.ac.at
2 Knowledge Management Division, Vienna University of Economics and Business, 1020 Vienna, Austria; birgit.fordinal@wu.ac.at
* Correspondence: alexander.kaiser@wu.ac.at; Tel.: +43-1-31336-5230
† These authors contributed equally to this work.

**Abstract:** The role of *spirituality in organizations* has received increasing attention over recent years. The purpose of this conceptual paper is to take up this shift and develop the foundations for an alternative approach to knowledge management: *Spiritual Knowledge Management*. A key question in spirituality concerns the unfolding of the identity (of an organization) or the self toward a "higher end" or purpose. We propose the concept of the *phronetically enacted self* (understood here both in an individual and an organizational sense) that helps us conceive of how this unfolding can be achieved in a thriving and sustainable manner. The self is conceptualized as a highly dynamic and emergent "entity" that is grounded in a continuous process of becoming and of transitions transforming a state of potentiality into a state of actuality and fulfillment. Insights from the theory of spirituality, enactive cognitive science, the theory of potentials/possibility studies, phronetic organizations, and resonance theory lead us to a novel understanding of knowledge-driven organizations embodying a spirituality-based and, as a consequence, (regenerative) sustainable approach. Finally, we will develop the basic characteristics and leverage points for transformative shifts toward sustainability in organizations.

**Keywords:** spiritual knowledge management; phronetically enacted self; phronesis; enactivism; knowledge management; innovation; future potentials; 4E cognition

## 1. Introduction

Over recent years, the role of *spirituality* in organizations and in Knowledge Management has been intensively studied and discussed in the literature. Topics, such as spiritual knowledge [1], spirituality in the workplace [2], responsible knowledge management [3], Spiritual Knowledge Management [4], to name a few, play an increasingly important role in scientific and practitioner conferences, scientific journals as well as in daily organizational practices. A common underlying question across these contributions and practices is: How can organizational life flourish, enabling individuals, the organization, and the world they inhabit to thrive and progress positively and in a sustainable manner?

Interestingly, it was Knowledge Management (KM) that took a pioneering role in this approach; many of these issues are addressed in Nonaka's concept of the *wise organization* [5] and the papers and discussions that followed [6,7]. Along these subjects, much attention has been paid to the topic of *purpose* in the last few years. Some argue that this more intense preoccupation with issues such as purpose and spirituality in the context of organizations is a consequence of the Covid pandemic and that, as a result, people are more concerned with such fundamental and existential questions compared to pre-pandemic times. For others, the professional reflection and discussion of topics such as calling, vocation, and spirituality is also a signal for a fundamental change and shift in the priorities of business development, business performance, business innovation, as well as a change in personal value systems. Especially in the field of spirituality, we can distinguish between more individual-centered

and more community-centered approaches. However, for the purpose of the argument in our paper, it would go far beyond the scope of this paper to take this distinction into account in more detail.

Discussions about spirituality and purpose are also in line not only with increased interest, but also with an alternative understanding of sustainability having a focus on the regenerative aspect (of organizations) [8]. From a knowledge management perspective, one could also argue that these developments call for a repositioning and redefinition of a future KM. Irrespective of this, the underlying phenomenon addressed in all of the topics briefly discussed here is a specific form of *transformation*. Thus, this paper explores the concept of transformation, which involves shifting a current state or situation toward a future state that is more advanced, intricate, authentic, and–from a normative perspective–a state better aligned and "in resonance" with what is considered ideal or "ought to be". This transformation can take various forms, such as evolving an ordinary organization into a wise organization [5], or the transformation of a non-sustainable organization into a sustainable and regenerative one, or guiding an individual from being in a state of potentiality to being and living in actuality. Central to this exploration is the transformation of both organizations and individuals toward fully embracing their inherent calling and vocation. Importantly, these transformations are purposefully initiated, aiming for profound, substantial, and meaningful change that is intentionally triggered.

Such a deep and existential transformation and ongoing development are key for organizations as well as for the individual to be successful and at the same time fulfilling their purpose and thriving. However, such a transformation will only be sustainable if it also takes into account a person's self (including their social and cultural background) and—in an appropriate manner—includes both the organizational level and the flourishing of their environment.

*Structure of the Paper*

In this paper, we will show that this also poses a knowledge management (KM) problem, as essential types of knowledge must flow between organizations, their employees, and the environment and ecosystem they are embedded in. It spans individual and organizational KM, as shown by by [9]. From a spiritual KM perspective, this flow of knowledge concerns the unfolding of the identity (of an organization) or the self toward a "higher end" or purpose. In order to achieve a better understanding of these processes, we introduce the concept of the *phronetically enacted self* (understood here both in an individual and an organizational sense) that helps us conceive of how this unfolding can be achieved in a thriving and sustainable manner. In our context, the self is conceptualized as a highly dynamic and emergent "entity" (or better, as a process) that is grounded in a continuous process of becoming and transitions from a state of potentiality into a state of actuality and fulfillment. Insights from the theory of spirituality, enactive cognitive science, the theory of potentials/possibility studies, the phronetic organization, and resonance theory lead us to a novel understanding of knowledge-driven organizations embodying a spirituality-based and, as a consequence, (regenerative) sustainable approach. Finally, we will develop the basic characteristics and leverage points for transformative shifts toward sustainability in organizations. This paper is organized around the following research questions: What is the phronetically enacted self? How is it related to Spiritual Knowledge Management? Why is it important for sustainability? As mentioned above, it is important to note that, in this paper, we first and foremost take an individual perspective on the phenomena we are going to investigate. However, we will also address the transfer to the organizational level to some extent, as far as this is possible and useful in this phase of our research.

## 2. Knowledge Management in Flux

KM has been undergoing fundamental transformations in recent decades [10]. In the last years, new and emerging topics such as Responsible Knowledge Management [3], KM in purpose-driven organizations [7], the wise company and phronetic leadership [5,11]

became key concepts in KM and are gaining traction in research and practice. Following Alvesson and Sandberg [12] concept of the "art of phenomena construction", we suggest taking these developments seriously and suggest to go beyond existing approaches in KM by introducing a novel concept. "Phenomenon construction" in research is understood as a *response* to what already can be observed in the theory and, more importantly, practice of KM. The goal of this paper is to introduce the concept of *Spiritual Knowledge Management* that has recently been proposed and has triggered interesting discussions [4].

### 2.1. Spiritual Knowledge Management

The main idea of Spiritual Knowledge Management (SpKM) is to focus on the development and becoming of individuals and organizations transitioning from the current self (or state) to a future self that is different and in some way more developed and unfolded than the current state as a result of a process of *self-realization* and *transformation*. Figure 1 shows this development schematically and highly simplified, since it rarely happens in such a linear and straightforward manner.

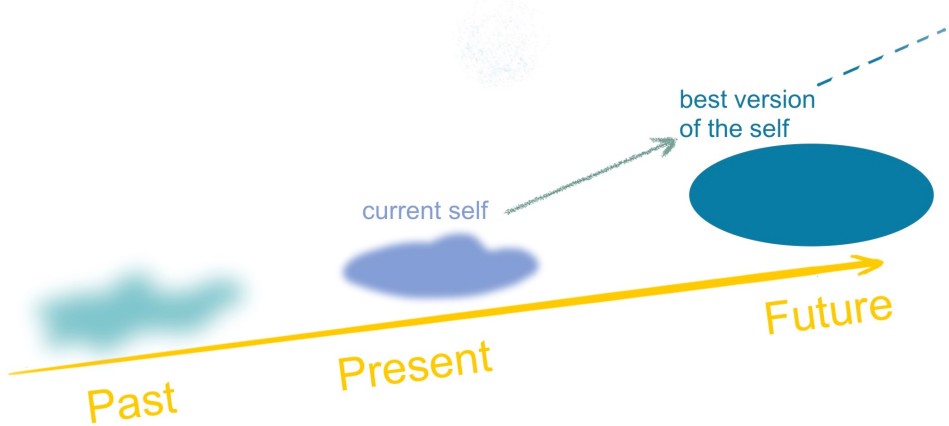

**Figure 1.** Process toward the unfolding of a future self.

As can be seen, a step-wise development takes place that can be understood as a transformation from the current version of a person's (or organization's) self to the *future version* of her-/him-/itself. At this point, it should be noted that not all types of spirituality can be adequately described as a process of personal growth. In Buddhist traditions, for instance, the concept of non-self is central to spirituality. In contrast, from a Christian tradition, the importance of the unfolding and development of the self is central, as every human being is seen as an image of God and the realization of this image is very strongly connected with the unfolding and development of the self. For the simplicity of the argument, we will focus on the person in the following sections; however, as will be shown below, these concepts can be extended and applied to the organizational domain as well.

What is the role of spirituality in this context? First of all, there is no single, widely agreed definition of spirituality. In a literature review based on around 100 articles and books, Tanyi concludes that spirituality is an inherent component of being human, and is subjective, intangible, and multidimensional and that it involves humans' search for meaning in life ([13], p. 500). One fundamental dimension and key element of spirituality, which can be found in almost all definitions and approaches of spirituality—even if they are sometimes quite different—is the *self* ([14], p. 1183) as spirituality often affords a setting for *self-exploration, personal growth, and the search for meaning and purpose in life* [13]. Especially inner transformation plays an important role in spirituality, as many spiritual traditions emphasize the importance of inner or personal growth, which can lead to personal transformation and to a more authentic and fulfilling life [15]. Moreover, the transcendence of the ego, which refers to identification with the individual self and its desires, is another

key factor of spirituality that is closely related to self-development. Hence, spirituality is a means of reaching a future version of the self.

As we are living in a highly dynamic and unpredictable world, this future version of the self cannot be (completely) known at the beginning of this path. Hence, this process has to be thought of as a process of *becoming* [16,17] and at the same time as a profound learning process. In the course of this learning process, knowledge about the very nature and shape of the future version of the self is created, and at the same time, this knowledge shapes and clarifies not only the future version of the self, but also the way how it can be achieves. Given the fact that the development of the self in such a sense of becoming is an essential aspect of spirituality, this approach to KM is consequently referred to as Spiritual Knowledge Management [4].

Going beyond the classic notions of knowledge, such as propositional knowledge or knowledge as "justified true belief" (e.g., [18]), we suggest following Nonaka and other scholars [5,19–21] in their more processual and functional perspective and characterize knowledge "as a generalized *capacity to act on the world*, as a model for reality, or as the *ability to set something in motion*" ([22], p. 1, emphasis by authors). This is important for our argument, as it stresses an understanding of knowledge as a *process* that is continuously changing and adapting to its unfolding environment. Accordingly, it is precisely the knowledge generated in this deep learning process that enables a person to *act* step by step on the way toward this future version of him- or herself. Consequently, changing existing and creating novel knowledge enables possible growth and the development of novel meaning for a person.

However, this development of the self toward a future self, which plays such an important role in Spiritual Knowledge Management, is nothing new and has been discussed by many other authors, although never in the context of KM. As will be shown in the following section, one can discover quite similar patterns when looking at approaches by some authors in the field.

### 2.2. The Transformation of the Self

In [4], we did a review of some prominent examples and concepts of how authors from very different fields and backgrounds have conceptualized different forms of the self and its becoming over time. In this section, we shortly reflect on and summarize this review.

- *Matthew Kelly*, a practitioner in both counseling and spirituality, advances the notion that by adeptly discerning our genuine needs, profound aspirations, and innate talents, we may uncover an "optimal rendition of ourselves" distinct from our authentic selves [23,24].
- *Richard Rohr*, drawing from a foundation in Catholic spirituality, delineates a distinction between the true self and the false self. According to Rohr, the true self is the facet of our identity that possesses a profound awareness of our essence and purpose. He underscores that the metamorphosis from the false self to the true self is substantially linked to the act of letting go [25,26].
- *Richard Boyatzis*, whose expertise lies in the domains of organizational behavior and coaching, introduces a dichotomy between the real self, which reflects an individual's current state, and the ideal self, signifying the potential that the individual can reach [27,28]. Grounded in this conceptualization, he has formulated the theory of Intentional Change, a five-step framework designed to facilitate individuals in realizing and maintaining desired transformations and reaching their objectives [29,30].
- *Stam et al.* [31] posit that the self-concept constitutes our perception of ourselves or our understanding of our own identity. Simultaneously, they establish a distinction between the presently experienced self and the possible self. The former is anchored in the present moment, accounting for current circumstances and constraints, while the latter is a facet of one's identity linked to the future, encapsulating not the genuine self (i.e., the present self) but rather the self that has the capacity for growth and development.

- 　*Claus Otto Scharmer*, whose expertise lies in leadership, transformation processes, change management, and action research, discerns a fundamental distinction between an individual's "current self" and the emergent future "(highest) Self", emphasized with a capital "S", signifying the fullest potential of an individual [32]. Within his Theory U framework, Scharmer introduces the concept of "presencing", a process through which individuals establish a connection with the wellspring of the most elevated future possibilities. This, in turn, facilitates the actualization of one's authentic self [33] and the generation of self-transcending knowledge that surpasses the limitations of the self [34].

In conclusion, even with very different topics and backgrounds (theology, consulting, leadership, change management, psychology, organizational behavior, coaching, etc.), the basic assumption of these approaches is actually very similar: A fulfilled and prosperous life is closely linked to a self that is unfolded or (self-)actualized in the very best way possible. However, we also can find several superlatives used in these descriptions: best self, ideal self, real self, true self, self in the sense of greatest potentials, etc. Looking closer, it turns out, however, that these superlative terms are not really adequate for two reasons: (a) they imply that, as soon as one has reached this superlative future self, no further development seems possible. This is neither plausible nor realistic, as both the world and its opportunities and constraints as well as the needs and possibilities of a person or of an organization continuously change and develop further and are highly specific. (b) Moreover, it is not entirely clear what "best", "ideal", "true", etc. mean in various contexts and will vary considerably for each individual or entity.

As we will develop in the sections to come, we suggest using the term *phronetically enacted self* instead, as it offers a perspective/orientation, direction as well as a dynamical aspect toward what could be meant by "best".

## 3. What Is the Phronetically Enacted Self?

To find an answer to what we mean by the "phronetically enacted self", we will first address the question of what it means to select adequate or "right" potentials by making use of the concept of phronesis. In a next step, we will show that the self is a highly dynamic and emergent entity that is grounded in a continuous process of becoming and transitions from a state of potentiality into a state of actuality and fulfillment. We will present the ontological foundations of this transition (i.e., the concepts of actuality/actuals and potentials). Finally, we will discuss how potentials become actuals through a process of enactment by making use of the enactivist approach from cognitive science.

### 3.1. Phronesis as Key Ingredient of the Phronetically Enacted Self

The concept of *phronesis* dates back to Aristotle and can be defined as "doing the right thing, in the right way, and at the right time" ([35], p. 113) or "the capacity to put into action the most appropriate behavior, taking into account what is known (knowledge) and what does the most good (ethical and societal considerations)" ([36], p. 1250). In other words, it is the "experiential knowledge, embedded in character, used by individuals to determine and follow courses of intentional action" ([37], p. 92). Phronesis has been gaining popularity in various fields in recent years [7,38–41] and can be seen as an intellectual virtue; it is generally understood as the ability to determine and undertake the best action in a specific situation to serve the common good [42]. Emphasizing this direction in both individual and corporate activity recognizes the relatedness of business (and personal life) with society and the (natural) environment, and leads to acting responsibly toward humans and other entities [43]. As Kragulj [7] points out, phronesis can be seen as a third type of knowledge besides tacit and explicit knowledge: it is action-oriented and encompasses value judgment and—from a resource-based perspective—it provides an organization with the resources necessary to act wisely. Phronesis is also often referred to as practical wisdom emphasizing the aspect of the capacity to act in the interest of the *common good*.

For our reasoning in this paper, the concept of phronesis is essential, since it is not about an enactment of the self in an "arbitrary" or "selfish" direction, but about an enactment and realization of the self in the direction of a phronetic one—i.e., having the common good in mind.

### 3.2. Dynamics and Development of the Self—From Potentials to Actualization

As we have seen above, it seems to be critical to place the self (or, in the context of organizations, its identity and purpose) at the center of our investigations. It is the carrier and the core of spirituality as well as the driver for any kind of (spiritual) development. Spirituality is intimately linked with the self as it is about its ultimate meaning, beliefs, values, and purpose in life; it concerns the person as a whole, their identity as well as the relationship to him-/herself and their social and non-social environment. Furthermore, we have seen that spirituality is a key driver for the evolution and development of the self toward a fulfilled and thriving life. Some refer to it as the "optimal being" ([13], p. 506), as the emerging "highest Self" [33], the "best-version-of-myself" [24], etc.

As previously noted, the authors maintain a degree of skepticism toward the superlative language employed in these descriptions (e.g., "best", "highest", etc.). As will become clear later, we prefer talking about the *phronetically enacted self.* The Aristotelian concept of *phronesis* (or practical wisdom) is in line with the approach to spirituality understood as a process of "becoming fully human". When one reflects on the underlying concepts, there is something that these notions have in common: First, the self is not static. Rather, it is highly dynamic and constantly changing; it seems that spirituality is the motor (or at least one of the drivers) of these changes and gives them some kind of orientation and direction. Second, as a consequence of this dynamics, in terms of the notions of a "fulfilled life", "the person he/she should be", or "becoming fully human" [44,45], one can identify a movement or transition from a state of unfulfillment or a state of "incompleteness" toward a state of fulfillment, or from a state of not being actualized (or being in a state of potentiality) to a state of being actualized.

In the sections to come, we will focus specifically on this transition and on the underlying ontological assumptions, as they turn out to be key to understanding what we mean by "phronetically enacted self". In this context, the concepts of potential and enactment play a central role.

### 3.3. Ontological Foundations—Actuals and Potentials

Reflecting on the development of the self, one will discover that, although having some kind of stability is expressed in its identity or purpose, the self is in a constant process of unfolding and becoming. "Like the world, human life is a venture, a series of risks, that is radically open to an indefinite future without a certain conclusion" ([46], p. 22). E. Bloch [47] describes the world as being an experiment (what he refers to as "experimentum mundi") that is not only in a process of perpetual unfolding, but also in an unfinished or incomplete state. This implies that the world in general, and the self (or an organization's identity) in particular, is in an open-ended process toward what is "*not yet*". However, due to its current determination, identity, and purpose, as well as its history ("path dependency") this open-endedness is constrained. This implies that one can only have an approximate idea of where this process of becoming is leading (in the present moment). Nevertheless, the self's purpose might change as well over time; however, this usually happens at a very slow pace.

Hence, future states of the self are concerned with what it is "not yet". As we have seen, although they are categorically open [48], they are partly determined by the directedness toward the not-yet. This implies that in the process of unfolding novel qualities, attitudes, meaning, behaviors, etc., and sometimes even a new purpose or goals might emerge and/or be revealed. In other words, there is something present in the present moment that is hidden, or, as Poli refers to it, that is *latent* [49], something that is not "directly visible" in the moment and that has "not yet" been realized. "The difference between

being hidden and being latent can be clarified as follows: hidden components are there, waiting for proper triggers to activate them. On the other hand, latent components do not exist at all in the entity's actual state... Hidden and open components interact with each other. They form the entity's space of possibilities... The whole of the entity therefore comprises both tendencies and latencies, possibilities and potentialities" ([48], p. 77f). An ontology/epistemology of "not-yet" or of potentialities is necessary in order to make sense of this seemingly paradoxical situation (see also Glaveanu's approach of possibility studies [50,51]: Although the not-yet is not (yet) directly perceivable, it is there; it is brought to light or into actuality in the process of unfolding by following its own dynamics, by interacting with its environment, or by creating and making use of its niche.

Spirituality as Bringing the Self's Not-Yet into Actuality

Spirituality has a lot to do with this development of the self's not-yet to its actuality: Even though spirituality as such cannot be grasped so easily and there are of course views of spirituality that are quite different to our approach, for us spirituality is meant as being inseparable from the deepest purpose/meaning, the self and its growth (see our explanations above). Therefore we have to ask ourselves how is it possible that the self may "encounter" its not-yet (future) purpose. In other words, how can it reach its self-actualization by co-becoming with its future? Ontologically speaking, we are confronted with the question of how to deal with an unknown future and how one can make use of it (in the sense of futures literacies [52–56]) in order to transform what the self *actually is* in the present moment into what it *potentially could become* in the future. In other words, how the self can *enact its (full) potential*.

The concept of potential/potentiality can be found already in Aristotle's Metaphysics [57], where he introduces the distinction between *potential(-ity)* and *actuality/actual(s)*. His understanding of reality is based on the idea that every being/object or phenomenon—although having some determination/identity—can be considered to be in a process of becoming—has in itself a latent tendency toward the realization of its potential leading to its (emergent) actuality, telos, purpose, or "entelecheia". The phenomenon, thus, is not fully determined in its future development in the sense of not being completely predictable at any specific point in time), it is unpretstatable [58–61]. The interesting point is that—though distinct—*both actuals and potentials are present in every entity*. In other words, every entity is both in a state of actuality and potentiality at any given moment, and through their interaction (with the world and its own dynamics) actuality continuously unfolds. "Becoming" then means that there is going on a constant transformation or unfolding of potentials as a form of *enactment/enacting*: actuals transition into new/changed actuals by realizing or enacting the entity's potentials. As mentioned earlier, this can happen through following the internal working and dynamics of the entity and/or by being influenced or impacted by external factors in interacting with the environment.

Comparing actuals and potentials, the latter are found to have different qualities both from an ontological and epistemological perspective. While actuals are "real" in the sense of being already existing and observable, potentials are not directly observable and—as we have seen above—in a state of not-yet or potentiality. Actuals, thus, exist already and are open to being changed or transformed by the realm of possibilities. Possibles or potentials are open to develop or realize in various ways and directions that are partially intrinsic to the entity and partially dependent on environmental stimuli, influences, or changes. They exist in the realm of possibilities/possibles/potentia [57] or "adjacent possibles" [58,59,62–64] as something that is already there, but is latent or hidden [48,49], as a kind of disposition for the emergence of actuality or as a niche to be actualized [65,66]. While actuals are in the present, potentials always point toward the future. They are intrinsically about future states or processes in the sense of being "unrealized potentialities that are latent in the present, and the signs and foreshadowings that indicate the tendency of the direction and movement of the present into the future." ([46], p. 16) Hence, as pointed out in Aristotle's Nicomachean ethics [67], apart from virtues, the dynamics and development

of the self are driven by this future-oriented transition from its unrealized potentials to actuals leading the self into its own future.

### 3.4. Enacting as a Process of Transition from Potentials to Actuals

After having introduced the notion of potentials and actuals, this section addresses the issue of *how these potentials are transformed into actuals* in the development of the self. We will draw on the approach of *enactivism* from cognitive science [68–76] and show that this transition is realized as a process of bringing forth and *enacting* a cognitive system's internal and external world by interacting with and acting in the world.

What are the theoretical foundations of such an enactive perspective on the development of the self? Enactivism is a relatively recent development in cognitive science that is part of the so-called 4E-approaches to cognition [72,76]: they claim that cognition is embodied, embedded, extended, and enacted. The enactivist approach was originally developed by Varela et al. [75] and is based on the assumption that every cognitive system has to be understood as a *living system* that is embedded in its environment and finds itself in a precarious state of its survival. Cognition is considered to be at the service of the cognitive system's becoming and survival (on all levels) by making sense of and acting in the world [68,69,77,78]. Its cognitive capabilities regulate the cognitive system's interactions with its (internal and external) environment and, by doing so, it tries to sustain its state of being alive. Above that, by regulating its interactions, the cognitive system *creates its meaning and maintains its identity* through its autonomy, autopoietic organization, and (structural) coupling to the environment [69,74,79,80]. Hence, "a cognitive agent is an autonomous system, that is, an operationally closed, self-organizing network of components that dynamically connect to each other in multiple ways. As organisms enact their autonomy, they establish patterns of correlation between movement and sensory stimulation that simultaneously distinguish the agent from its environment and identify meaningful relations within it. Cognitive structures thus emerge from enaction" ([81], p. 2).

This is opposed to classical approaches to cognition, such as cognitivism, that take a primarily representational perspective to cognition: the idea is that cognition "represents" or "is about" the world (by making use of knowledge structures) and operates on these representations (e.g., by applying rules to propositional knowledge) [76,82–84]. This is in contrast to enactivism, which takes a radically *action- and interaction-oriented* position: the purpose of cognition is to generate meaningful behavior, act in the world, and produce its own cognition and "things". The goal is to ensure the organism's survival by making sense of the world and bringing meaning to its internal and external world. In his material engagement approach, Malafouris [85,86] refers to these activities as "*thinging*" as opposed to "thinking"; in this sense, "thinging denotes the kind of thinking we do primarily with and through things. For the material engagement approach witness and throughness takes precedence over aboutness" ([86], p. 7f).

Hence, "a cognitive being's world is not a pre-specified, external realm, represented internally by its brain, but a relational domain enacted and brought forth by that being's autonomous agency and mode of coupling with the environment" ([87], p. 13). This is achieved in a process of interaction between the cognitive system and its environment, mutually adapting, (co-)creating, sense-making, and enacting on each other. In other words, "organisms regulate their interactions with the world in such a way that they transform the world into a place of salience, meaning, and value... This transformation of the world into an environment happens through the organism's sense-making activity" ([74], p. 25). Coming back to our discussion from above, *enacting* both the self and the world by interacting with and making sense of the world is the process standing behind transforming potentials into actuals.

### 3.5. (Positive) Resonance—Towards a Phronetically Enacted Self

This brings us to the question, of what it means for the self to enact itself in the sense of bringing its/her/his potentiality/"not-yet" into actuality; more specifically, as we intend to replace the notion of the "best version of the self" or "highest self", we have to come up with an alternative concept that covers both the dynamical aspect and the need for reaching some kind of completeness or fulfillment as a future state of the self or of an organization (see above). We propose to look at the enactment of the self through the lens of *(positive) resonance* in the process of interaction and coupling between a cognitive system (or an organization) and its environment.

The notion of "best version of the self" or similar approaches suggests that there is some external measure or (given and stable) goal state the self has to reach. This leads to a situation where the goal is to find some kind of "fit" between the self and the environment. This implies that we run into the trap of looking at and reducing this relationship to a problem-solving task. Such a strategy is neither future-oriented nor does it take into consideration the full potential of the autonomy and dynamics of the self as well as the self's capacity to shape and influence its environment. Such a perspective is rather reactive and mostly driven by past experiences as well as by a mindset of adaptation and optimization. In a way it is opportunistic as it "runs behind" what is happening or already has happened in the environment/world and tries to adapt to it by hoping to find some kind of fulfillment by fitting into it.

While this relationship of a fit between the self and its environment is a rather static and mechanistic concept, we propose replacing it with the concept of *(positive) resonance* aiming at establishing a *dynamic relationship* of *interaction, emergence, correspondence* [88], and *co-becoming* [89,90] with the world. As discussed earlier in our exploration of enactivism, this implies a form of *dynamic coupling* respecting the autonomy and purpose of the participating systems; these systems together form an *ecosystem* encompassing the self and the environment including the systemic and ecological context. The components of the newly emerged system engage in mutual co-enactment by providing inputs, possibilities, constraints, and value. Collectively, this ecosystem strives to proactively shape and co-create a niche that facilitates a shared thriving future [63,65,89–91]. This joint endeavor can be seen as a dynamic process of positive resonance and flourishing, where the system continually finds itself in a state of dynamic change.

Our understanding of resonance is based on H.Rosa's [92] concept of resonance as a sociology of world relations. As we have seen in our discussion about enactivism, the self–world relation is our focus if we are interested in how a cognitive system enacts its purpose. Rosa defines resonance as an *emergent* phenomenon characterizing the human–world relation(s): resonance emerges when a human and the world "meet" and engage in a process of *(mutual) transformation*.

Rosa's approach to resonance rests upon two fundamental assumptions: its inherent relational nature and the innate resonance-seeking tendency of human beings. These principles manifest themselves across various aspects of our lives: in our epistemological relationship to the world, in our social relationships, in how our internal needs and drives resonate with what the world offers us (e.g., affordances [93,94], opportunities, constraints), etc. Although the term might suggest an acoustic phenomenon, we must not reduce resonance to its purely mechanical dimension. Rather, Rosa suggests that resonance always involves a considerable level of unpredictability, self-organization, and emergence due to the *autonomy* of the entities involved in activities of resonance. As we have seen, this autonomy is rooted in the agency of the participating systems. This point is of importance as it helps us understand that resonance is primarily about a relationship of (mutual) *response*, rather than echo or (passive and predetermined) "reaction". Resonance, then, involves (a) respecting the participating systems' *autonomy* and (b) creating, participating, and engaging in a relationship of *mutually responding* to each other. Thereby, we are building a meaningful and transformative rapport between agents and their environment leading—in the best case—to their flourishing.

This gives rise to several properties and implications that are interesting with regard to our question of enacting the self (or an organization) by interacting with the environment: (a) Living and social systems exhibit *self-oscillations* (Eigenschwingungen); being in resonance with oneself is a prerequisite for engaging in resonance with the environment (compare, for instance, the homeostatic equilibrium found in autopoietic systems [95–97]). (b) Only if a system has found its self-resonance, it will be able to articulate its "own voice" to another system. Both systems (i.e., the self and its environment) follow their own dynamics and are in a state of self-oscillation, while, in their interaction, they are exposed to the "vibrations" of the respective other system(s). In this "relation of resonance, these entities 'speak with their own voice', thus not only affirming their relationality and reciprocity but also retaining a substantial degree of independence" ([98], p. 312f). It is in this process of listening and responding to each other that all involved systems *maintain their autonomy* and "speak with their own voice" and at the same time are *open to being transformed* by what they perceive and "hear". This subtle interplay between autonomy and being influenced by the environment is the foundation for a dynamic development of the self toward its actualization. (c) While in an echo-relationship, we find a kind of re-action to or inter-action with the external world, a self resonating system together with its environment engages in a process of *co(r)-respondence* [88,99] potentially leading to co-enacting each other's future potentials and joint relationship. (d) This, in turn, may lead to one's self-actualization and finding one's purpose. Engaging actively in resonance means being in resonance with oneself (in the sense of one's purpose) and with one's environment. Therefore, it is necessary to create and establish such "spaces of resonance" [92] that can function as enabling spaces [100,101] fostering and supporting practices of obtaining positive resonance.

*3.6. The Phronetically Enacted Self as Being in a State of Resonance with Emergent Future Potentials and Actualizing Them in a Thriving Manner*

Resonance is not only about describing the relationship between two or more systems but also has a normative character: it is about how things should/could be. In that sense, resonance may serve as a guiding principle and measure for a "good and purposeful life" in the sense of, for instance, Aristotle's [35] concept of eudaimonia. Therefore, resonance always points toward the future and toward realizing future potentials as forms of positive resonance. On the one hand, resonance has a highly dynamic, emergent, and open-ended character, on the other hand, resonance is always about *desired states* in the future. It is about possible future states (of the self or of an organization) that "want" to emerge with different probabilities of realization. As a consequence, in order to co-create and enact such a meaningful and thriving future, we have to learn how to make use of these emerging potentials and how to "learn from the future as it emerges" [33].

What do the considerations in the above sections mean for our understanding of the proposed concept of the phronetically enacted self? As has been discussed above, the concept of *phronesis* plays a central role in this approach. We want to build on Bachmann et al. [102] comprehensive and widely accepted characterization of phronesis to better understand how to realize future potentials in an individual as well as in an organization in a thriving and flourishing manner. Bachmann suggests understanding phronesis as practical wisdom that "improves managerial reasoning, decision making, and acting, concurrently (1) integrating and balancing several, often competing interests, rationalities, emotions, challenges, and contexts, (2) orientating toward normative guidance of human flourishing, (3) considering the indispensable sociality of every human being as well as (4) today's multilayered diversity in life and society, (5) acting appropriately and authentically in a self-aware manner, (6) rediscovering transmitted cultural and spiritual heritage, (7) being aware of the incompleteness of human existence and humble in the face of one's own achievements and capabilities, and (8) targeting always realization in practice" ([102], p. 162). What are the implications of this perspective on phronesis for our concept of the phronetically enacted self?

1.  First, as previously indicated, the term "self" encompasses both the individual self and the "self of an organization", referring to its identity and/or purpose. The self, although having a relatively stable core, is a dynamic entity that is in a constant process of unfolding.

2.  As shown by Bachmann [102], phronesis is consistently oriented toward human flourishing and the pursuit of the common good. Hence, phronesis is always directed toward living both in harmony with society and the environment as well as with oneself (compare also Nonaka's concept of a wise organization [5]). This can only be achieved by adopting a humble stance in the face of the vulnerability, fragility, and unpredictability inherent in today's world.

3.  We have characterized the unfolding of the self as a process of *(self) realization*, i.e., as a process of becoming that transforms (future) potentials into actuals. The potentials' realization depends both on the inner dispositions and dynamics of the self and on interactions with and interventions from the environment. These processes are intrinsically future-oriented. This implies that we are facing the issue of how future potentials are (a) *anticipated*, (b) chosen, (c) realized in a process of *enaction*, and (d) possibly adapted. In contrast to a prediction that is based on knowledge from the past (this is in the regime of the efficient cause), anticipation always means that we must acknowledge that "future states may determine present changes of state" ([103], p. 770). For the same reason, v.Foerster observes that, from a complex system's perspective, it is not sufficient to draw on the efficient cause to understand the unfolding of a system, but that "its cause lies in the future" ([104], p. 230). As he shows, this approach is based on the concept of *final cause* going back as far as Aristotle [57].

4.  Being the "why?", the final cause is closely related to the *purpose* of the self or an organization. This implies that the purpose lies in the future and cannot be realized with a planning or controlling attitude. Rather, it is about being open and receptive to the emerging purpose, so that we can sense and identify [33] this purpose in the future potentials. Instead of "making" we have to adopt an attitude of "*being attracted*" by this future purpose and of having the openness for *being transformed* by it.

5.  The main point of such a process of (self-)enactment or (self-)actualization is to *give up control* and engage in a relationship of correspondence [88,99] and *resonance* [92] with the unfolding world. This reflects what Bachmann et al. [102] mean by that we have to be aware of the incompleteness and unfulfilled state of our human existence and, as a consequence, that we have to assume a more humble position.

So what does the "phronetic" refer to in the concept of the phronetically enacted self? What we are proposing here is a *wisdom-based* perspective, similarly as in Nonaka's wise company approach [5,11]. Such a wisdom-based perspective is strongly related to a spiritual perspective, as they share the same viewpoint [105]. However, we suggest going one step further by introducing the concepts of (future) potentials and how they are transformed into actuals. As we have seen, this is achieved through enacting the individual's (or organization's) internal and external world by interacting with it, by shaping it as well as by being shaped by it. In this context, the concept of resonance [92] turned out to be central. Engaging in a relationship of resonance with one's environment implies both openness to what wants to emerge in the world and at the same time sustaining one's identity. More specifically, both systems, the self (or the organization) *and* the world engage in a process of *co-creation* and co-shaping each other by listening to each other and to the mutual emergence of potentials in their space of interaction. It is by cultivating this interface of emerging future potentials that enables dynamics in which both systems can thrive and find a form of self-actualization or fulfillment. Phronesis is exactly about this situation-awareness of doing the right thing in the right moment and coming up with the appropriate behavior by taking into account all available knowledge aiming at doing the most good and bringing the common good to life ([36], p. 1250) (see our discussion above). Phronetic behavior is thus reflected in being in this state of resonance, in giving up control

and being open to letting oneself be transformed by an unfolding world. However, this resonance is not only about being receptive and transformed, but also about a sensitive intervention and transformation of the world sustaining this state of resonance.

This shaping the world and being shaped by the world in a resonating manner not only leads to selecting the "right" potentials for enacting a joint thriving ecosystem, but also sheds a new light on our understanding of sustainability and sustainable (organizational) behavior.

## 4. The Phronetically Enacted Self and Sustainable Organizations

Sustainability has become a key challenge for our society and organizations in recent decades; it has turned out as critical in today's world to meet the needs of the present without compromising the ability of future generations to meet their own needs. It is crucial for safeguarding the environment and climate, conserving resources, and ensuring a prosperous and resilient future for all. Framing the phronetically enacted self as an entity emerging in constant interaction and resonance with itself and its environment leads to an alternative understanding of sustainability. This is important as both individuals and organizations that are following such a phronetically enacted strategy are at the heart of sustainable behavior. In this section, we will show how the phronetically enacted self can serve as a foundation for enabling sustainable behavior on an individual, organizational, and societal level and how it can implement an advanced approach to sustainability. In doing so, we adopt a meta-perspective on the entire approach to sustainability and deliberately refrain from differentiating between the pillars of sustainability (environmental, social, economic, cultural) in order to focus on the underlying mindset.

### 4.1. Regenerative Sustainability—From Sustaining to Thriving

The field of sustainability has undergone significant changes in recent decades and is on the threshold of a new paradigm [8,106,107]. It has become apparent that it is no longer sufficient to strive for a minimum level of human well-being within planetary boundaries (cf. "Brundtland Report"). The prevailing focus on individualization and independence, coupled with consumerism or "the story of more" [108] are the accompanying manifestations of a mechanistic worldview in which humans perceive themselves as separate from nature. Conventional and current approaches to sustainability have "focused on the external world of socio-economic structures, governance dynamics, economic incentives and technology" ([109], p. 2). By mostly looking at symptoms rather than causes of unsustainability, the attention has been on shallow leverage points like finding more efficient technologies or policy changes.

There is a growing recognition that the underlying mindsets and attitudes are the root of the problem, but they also present an opportunity for solutions [8,110–112]. *Regenerative sustainability* represents a novel paradigm that encompasses a necessary shift to a holistic and systems worldview or as du Plessis puts it, "a shift from seeing the planet as a deterministic clockwork system in which humans are separate from nature to seeing it as a fundamentally interconnected, complex, living and adaptive social-ecological system that is constantly in flux" ([106], p. 12). Gibbons goes as far as saying that, "(r)egenerative sustainability sees humans and the rest of life as one autopoietic system in which developmental change processes manifest the unique essence and potential of each place or community. Regenerative sustainability's aspirational aim is to manifest thriving and flourishing living systems (i.e., complex adaptive systems) in the fully integrated individual-to-global system" ([8], p. 3). In this sense, the concept of regenerative sustainability goes hand in hand with the approach of a phronetically enacted self as dynamic, emergent, and characterized by positive resonance.

In a similar way, Scharmer proposes a needed shift from ego-system to eco-system economies and states that, "the evolution and complexity of the real economy call for an evolution of our awareness from 1.0 (habitual), 2.0 (caring about the well-being of myself), and 3.0 (caring about the well-being of myself and some of my direct stakeholders) to

4.0 (caring about the well-being of myself, all stakeholders, and the whole eco-system)" ([113], p. 198).

### *4.2. Inner Sustainability, Outer Sustainability*

As we have stated, a phronetically enacted self is closely linked to the thrivability of a person and consequently to the thrivability of whole systems, such as organizations. Along these lines, there is a growing awareness in sustainability science that the existing external orientation on structures and technical challenges needs to be complemented with an internal focus in order to open up the possibility for a cultural shift in mindsets and profound transformation [8,111,114]. For instance, the perspective of climate change as a social problem [115] also takes the interconnections with many other social concerns (e.g., poverty, health, inequality) into account. The internal aspects of sustainability are often described as underlying and therefore unobservable aspects such as beliefs, values, worldviews, implicit assumptions, and paradigms. Emotions, values, desires, and goals also play an important role [8,116,117]. The external dimension of sustainability is manifest in observable aspects such as policies and governance structures.

Several scholars [112,114] have focused on identifying the underlying patterns and transformative qualities that are considered to be determinants for internal change to effectuate external change toward sustainability. Wamsler et al. identified the following five clusters of transformative qualities/capacities: "Awareness—the ability to meet situations, people, others and one's own thoughts and feelings with openness, presence and acceptance. Connection—the ability and desire to see and meet oneself, others and the world with care, humility and integrity, from a place of empathy and compassion. Insight—the ability to see, understand and bring in more perspectives for a broader, relational understanding of oneself, others and the whole. Purpose—the ability to navigate oneself through the world, based on insights into what is important (intrinsic, universal values). Agency—the ability to see and understand broader and deeper patterns and our own role in the world in this regard, and to have the intention, optimism and courage to act on it" ([114], p. 8). Similarly, Rimanoczy [112] describes a sustainability mindset that encompasses the following four areas: ecological worldview, systems perspective, emotional intelligence, and spiritual intelligence. Reflection, self-awareness, mindfulness, and purpose play a crucial role in realizing this different mindset and in bridging the gap from knowledge to action.

From our perspective, the described qualities correspond with the proposed processes of self-actualization in a phronetically enacted self; in other words, a phronetically enacted self can be seen as the realisation of the interplay of inner change and outer change.

## 5. Towards Spiritual Knowledge Management

In the last part of our paper, we outline and discuss how the concepts of the phronetically enacted self as well as the considerations in the previous sections can point us in the direction of developing the perspective of *Spiritual Knowledge Management*. Spirituality is connected to a perspective of wisdom that has been developed in the previous sections. It concerns the relationship between both an individual and an organization and their/its environment.

### *5.1. Leverage Points for Sustainable Transformation*

The concept of leverage points, as proposed by Meadows [118,119], refers to specific areas or elements within a complex system where small changes or interventions can lead to significant and transformative shifts in the system's behavior. These leverage points are points of strategic intervention that have the potential to generate substantial and lasting impacts that are key factors for spiritual development. Meadows identified various levels of leverage points, ranging from shallow leverage points such as parameters and feedback loops to deeper leverage points which include the structure of systems and its rules. Deep leverage points that result in high-level changes concern the area of intent and include goals and the mindset or paradigm out of which the system arises. The inner

dimension of sustainability having been discussed above corresponds to these deepest leverage points [8,120]. Understanding and effectively combining these leverage points can offer opportunities to catalyze systemic change and promote sustainable outcomes within complex systems [116].

The concept of a phronetically enacted self corresponds to the vision of internal development that, as a result, leads to practiced regenerative sustainability both at the individual, organizational and societal levels. Similarly Woiwode et al. point to the "interrelation of the self and societal change, of self-development and socio-cultural transformation in sustainable development" ([120], p. 845).

## 5.2. Wisdom and Sustainability

To link the individual to the organizational (and subsequently the societal) level and to support the connection between inner and outer sustainability it is also helpful to look at different forms of knowledge and knowing. Abson et al. state, "the way knowledge is created, shared and used in society crucially influences transformation processes" [116].

The Aristotlean concept of phronesis, having been described above, has been recognized as essential for sustainability transformations in organizations [111,121–123]. Living in a VUCA world and looking at sustainability as a wicked problem [124] that needs to be dealt with on different levels, and demands new capabilities, skills, and knowledge [11]. Wise decision-making is one of the essential skills to deal with the current complex situation of a polycrisis, or to put it very simply: sustainable practices represent wise practices ([5,125], p. 622). It is also interesting to look at the linkage between wisdom and sustainability: both incorporate ethics and morality into organizational practices or as Intezari puts it, "in the midst of wisdom and sustainability is a strong emphasis on considering the well-being of oneself and others, which requires a self-transcendence approach to the human and surrounding environment as an integrated whole" ([125], p. 622).

Phronesis is a helpful approach within the organizational context as it "enables us to see situations holistically and to reduce the complexity of VUCA environments. It provides a robust and future-oriented heuristic that enables us to weigh alternatives and make decisions for the common good" [40]. Therefore creating "spaces of resonance", spaces that enable the development of a phronetically enacted self, serves as the fundamental basis for regenerative sustainability as well as for a path toward Spiritual Knowledge Management.

## 5.3. Spirituality and Knowledge Management

As we have seen, there is no single, broadly agreed definition of spirituality [13]. However, what can be found in most definitions is that it consistently involves the "self" as a fundamental element, serving as a platform for self-exploration, personal growth, and the pursuit of life's meaning and purpose. Inner transformation, central in many spiritual traditions, leads to a more authentic and fulfilling life while transcending the ego is another key facet closely related to self-development.

However, not only is the self an essential factor of spirituality, but also the whole aspect of *resonance*, as spirituality, covers alignment, harmony, and connection with oneself, others, the environment, and, ultimately, with the universe [126]. As we have seen above, being in resonance with oneself, or in other words, being in resonance with "one's self", is a prerequisite for enabling the becoming of a fully unfolded and enacted self as well as for any sustainable action.

As mentioned earlier, all this also represents a knowledge management problem, both at the individual and the organizational level: essential types of knowledge must flow between organizations and their employees—in both directions—in order for work to become and remain sustainable, meaningful, fulfilling, and productive not only for both sides, but also for their users in the market. This fundamental knowledge flow will only be possible if there first exists a knowledge flow on the individual level, which has to take place between the person and the self of the person. In order to manage these knowledge flows, it seems essential to enable the creation as well as the transformation

of these different types of knowledge. All these aspects, to list just a few examples here, are essential elements of the *Spiritual Knowledge Management*. Therefore we believe that the approach of Spiritual Knowledge Management as a well-structured deep-learning process toward the phronetically enacted self is a promising dimension of the Knowledge Management of the future.

Of course, the Spiritual Knowledge Management approach is currently defined only in an initial, still very basic manner. An initial research agenda has been defined in order to continuously develop this field [4]. One item on this research agenda is the application and adaptation of systemic coaching methods for Spiritual Knowledge Management to enhance the deep learning process and create the essential types of knowledge. For this purpose, in particular, the method of coaching with compassion [127] and the method of vocation-coaching [128] have shown to be successful in practice. Both types of coaching can be seen as transformative learning processes [129,130] as they clearly facilitate the process of effecting change in a frame of reference [131]. Such a transformational learning process allows one to learn about the phronetically enacted self and the qualities that characterize and define that future version. Therefore, both coaching approaches can be described not only as transformational learning processes but also as also deep learning processes.

Future work in the field of Spiritual Knowledge Management will include, on the one hand, addressing the items on the research agenda step by step. On the other hand, for the reception of this new approach, it will be important to move forward the scientific discourse not only in terms of the overall approach of Spiritual Knowledge Management, but especially as regards the concept of the phronetically enacted self that has been proposed in this paper.

**Author Contributions:** Writing—original draft, M.F.P., A.K. and B.F. All authors have read and agreed to the published version of the manuscript.

**Funding:** This research received no external funding.

**Institutional Review Board Statement:** Not applicable.

**Informed Consent Statement:** Not applicable.

**Data Availability Statement:** Not applicable.

**Conflicts of Interest:** The authors declare no conflict of interest.

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
