# Peer review of "Enabling the Phronetically Enacted Self: A Path toward Spiritual Knowledge Management"

_sustainability, doi:10.3390/su151813957_

Round 1

Reviewer 1 Report

This article has a great deal of potential, but the concept of spirituality it presents is quite vague. According to the authors, the concept of self is essential to spirituality. They should specify which types of spirituality emphasise the importance of the self; for example, popularised western spirituality or new age spirituality? In addition, the article would benefit from a discussion or at least a mention of the relationship between spirituality and religion. Moreover, the connection between practical reason (phronesis) and spirituality remains ambiguous. I have the impression that the authors are juggling too many objects. I am not suggesting that they must necessarily choose between spirituality and phronesis, but the relationship between these two ideas must be clarified in some way. 

I have marked the attached file with my specific comments.

The text demonstrates commendable linguistic proficiency; it is coherent and notably free of typographical errors.

Author Response

Line #

Response

30

We have included a sentence in our paper that addresses the reviewer's comment.

44

Rewrote large parts of the paragraph & added a comment on the normative approach

48

We have added a paragraph at the beginning & end of the section "Structure of the paper" in which we point out that we first and foremost take an individual perspective in our paper and only touch the transfer to the organizational perspective marginally. However, the reviewer is absolutely right with his comment and we will then dedicate more attention to the organizational level in future work.

54

Added a short sentence

78

done

95

Thank you for this comment. We have added a short paragraph that addresses the reviewer's comment.

109

Added sentences why we went for this choice

112

Reformulated sentence

122

We have added a paragraph at the beginning & end of the section "Structure of the paper" in which we point out that we first and foremost take an individual perspective in our paper and only touch the transfer to the organizational perspective marginally.

125

The reviewer is absolutely right in his comment. However, this understanding of self is merely the view of Matthew Kelly, whom we quote here, and not ours. In fact, as a result of our work, we come to a conclusion very similar to the reviewer's comment.

159

Adapted text according to comment

197

We have modified this part of our paper so that it addresses the reviewer's comment.

208

Modified this statement so that it addresses the reviewer's comment.

255

We have modified this part of our paper so that it addresses the reviewer's comment.

289

Accommodated text according to reviewer’s comments

488

We have modified this part of our paper so that it addresses the reviewer's comment.

513

Thank you for this comment. We added a sentence at the end of the paragraph to clarify the perspective on sustainability.  The reviewer is absolutely right, that it would be a promising field for further research to look into the different thematic dimensions (and especially their interplay) within sustainability through the lens of spiritual knowledge management.

638

As reviewer suggested, we shifted this paragraph to the beginning of the paper and and rewrote it

Reviewer 2 Report

I think it's an interesting paper on spiritual knowledge management. It's a bit abstract, but worth a read. However, I have the following questions:

[lines 96-97] The authors argue that spiritual direction management at the individual level can be applied at the organizational level. However, I believe that the individual and organizational levels are different, meaning that what works at the personal level is not guaranteed to work at the organizational level. The possibility of individualistic error cannot be ruled out.

The authors use a variety of theories (the theory of spirituality, active cognitive science, the theory of potentials and possibility studies, phonetic organizations, and resonance theory) to explain the sustainability of spiritual knowledge management. However, the book lacks sufficient explanation of each theory, making it difficult to understand and somewhat distracting.

The current discussion of the implications appears somewhat cursory. It is important to consider what we can do with the core of this study. This means that the authors should provide more specific managerial implications.

Author Response

Line #

Response

96f

[lines 96-97] The authors argue that spiritual direction management at the individual level can be applied at the organizational level. However, I believe that the individual and organizational levels are different, meaning that what works at the personal level is not guaranteed to work at the organizational level. The possibility of individualistic error cannot be ruled out.

→ We have added a paragraph at the beginning & end of the section "Structure of the paper" in which we point out that we first and foremost take an individual perspective in our paper and only touch the transfer to the organizational perspective marginally.

X

The current discussion of the implications appears somewhat cursory. It is important to consider what we can do with the core of this study. This means that the authors should provide more specific managerial implications.

→ we have developed some implications in the sections 5.1 & 5.2

X

The authors use a variety of theories (the theory of spirituality, active cognitive science, the theory of potentials and possibility studies, phonetic organizations, and resonance theory) to explain the sustainability of spiritual knowledge management. However, the book lacks sufficient explanation of each theory, making it difficult to understand and somewhat distracting.

→ we tried to clarify the concepts/theories so that they become better accessible to the reader (see the boldface sections in the new version)

Reviewer 3 Report

This paper contributes to the evolving discourse on spirituality in organizations by proposing the concept of Spiritual Knowledge Management, exploring the phronetically enacted self, and offering a multi-theoretical framework for understanding and implementing sustainable organizational practices.

The paper introduces the concept of Spiritual Knowledge Management as an alternative approach to traditional knowledge management in organizations. It acknowledges the growing interest in spirituality within organizational contexts and proposes that a spiritual perspective can inform how knowledge is managed within organizations.

The paper proposes the notion of the "phronetically enacted self" in both individual and organizational senses. This concept refers to the process through which the identity and purpose of an individual or organization unfold towards a higher end or purpose. This conceptualization highlights the dynamic nature of the self and emphasizes the continuous process of transformation and transition from potentiality to actuality.

The authors draw insights from a diverse range of theoretical perspectives, including the theory of spirituality, enactive cognitive science, theory of potentials/possibility studies, phronetic organizations, and resonance theory. This interdisciplinary approach enriches the understanding of how spirituality and knowledge management intersect and contribute to the sustainable development of organizations.

By integrating insights from various theoretical frameworks, the paper proposes a novel understanding of organizations that are knowledge-driven while also embodying a spirituality-based perspective. This approach suggests that organizations can operate from a sense of purpose and higher meaning, leading to a regenerative and sustainable mode of functioning.

The paper concludes by outlining the fundamental characteristics and leverage points that can facilitate transformative shifts towards sustainability in organizations. These characteristics and leverage points likely encompass elements that promote a spiritual perspective, knowledge-driven practices, and resonance among members, all contributing to a more sustainable organizational culture.

In conclusion, the paper makes a notable contribution to scholarship by introducing the concept of Spiritual Knowledge Management, proposing the "phronetically enacted self" framework, and integrating various theoretical perspectives. Its interdisciplinary approach and potential for bridging the gap between spirituality and management could have a positive impact on both academic discussions and practical implementations in organizational contexts.

Great appreciation for the authors.

Author Response

  • Reviewer 3 did not make any negative remarks

Round 2

Reviewer 1 Report

The authors have adequately considered the critical remarks on the earlier version of their article. This has mostly been accomplished by appropriately restricting the scope of the article. However, even if the authors' chosen concept of spirituality is individual-centered, the context of the article appears to suggest that the study discussing the role of spirituality in organisations could benefit from a more community-oriented perspective. Perhaps the authors can do that in the future.